# Identifying multiple sclerosis subtypes using unsupervised machine learning and MRI data

Arman Eshaghi [1,2✉], Alexandra L. Young[2,3], Peter A. Wijeratne [2], Ferran Prados [1,2,4], Douglas L. Arnold[5], Sridar Narayanan[5], Charles R. G. Guttmann[6], Frederik Barkhof [1,2,7,8], Daniel C. Alexander [2], Alan J. Thompson [1], Declan Chard [1,9,10] & Olga Ciccarelli[1,9,10]

Multiple sclerosis (MS) can be divided into four phenotypes based on clinical evolution. The pathophysiological boundaries of these phenotypes are unclear, limiting treatment stratification. Machine learning can identify groups with similar features using multidimensional data. Here, to classify MS subtypes based on pathological features, we apply unsupervised machine learning to brain MRI scans acquired in previously published studies. We use a training dataset from 6322 MS patients to define MRI-based subtypes and an independent cohort of 3068 patients for validation. Based on the earliest abnormalities, we define MS subtypes as cortex-led, normal-appearing white matter-led, and lesion-led. People with the lesion-led subtype have the highest risk of confirmed disability progression (CDP) and the highest relapse rate. People with the lesion-led MS subtype show positive treatment response in selected clinical trials. Our findings suggest that MRI-based subtypes predict MS disability progression and response to treatment and may be used to define groups of patients in interventional trials.

[1] Queen Square Multiple Sclerosis Centre, Department of Neuroinflammation, UCL Queen Square Institute of Neurology, Faculty of Brain Sciences, University College London, London, UK. [2] Centre for Medical Image Computing (CMIC), Department of Computer Science, Faculty of Engineering Sciences, University College London, London, UK. [3] Department of Neuroimaging, Institute of Psychiatry, Psychology and Neuroscience, King's College London, London, UK. [4] e-Health Centre, Universitat Oberta de Catalunya, Barcelona, Spain. [5] McConnell Brain Imaging Centre, Montreal Neurological Institute, McGill University, Montreal, QC, Canada. [6] Center for Neurological Imaging, Brigham and Women's Hospital, Harvard Medical School, Boston, MA, USA. [7] VU University Medical Centre, Amsterdam, The Netherlands. [8] Department of Brain Repair and Rehabilitation, UCL Queen Square Institute of Neurology, Faculty of Brain Sciences, University College London, London, UK. [9] National Institute for Health Research University College London Hospitals, Biomedical Research Centre, London, UK. [10] These authors contributed equally: Declan Chard, Olga Ciccarelli. ✉email: a.eshaghi@ucl.ac.uk

There is a pressing need in neurology to define disease phenotypes based on their underpinning mechanisms, as an important step towards stratified medicine. The consequences of redefining disease subtypes based on biology rather than on clinical grounds alone are that clinical trials should be better able to recruit patients who are likely to benefit from the medication under investigation. New technologies, such as artificial intelligence and machine learning, can evaluate multidimensional data to identify groups with similar features. Such methods, applied to visible abnormalities on MRI scans, have great promise in classifying patients who share similar pathobiological mechanisms rather than common clinical features[1].

Multiple sclerosis (MS), which affects more than 2.8 million people globally, is primarily classified according to clinical symptoms rather than on well-defined pathological mechanisms[2]. Current practice divides MS into four phenotypes: clinically isolated syndrome (CIS), relapsing-remitting MS (RRMS), primary-progressive MS (PPMS) and secondary progressive MS (SPMS)[3]. Two descriptors underly these phenotypes: (i) disease activity, as evidenced by relapses or new activity on magnetic resonance imaging (MRI) and (ii) progression of disability[3]. Phenotypes and their descriptors are routinely used in clinical trials to select patients and to guide treatment assignment.

Imaging, immunologic, or pathologic investigations often show more similarities than differences across the MS clinical phenotypes: CIS patients may evolve into RRMS, and the majority of RRMS patients transition into SPMS over time[2,3]. The precise timing of these transitions is challenging to ascertain, because they are often based on the *subjective* recollection of symptoms and interpretation of signs. SPMS and PPMS share many MRI features and pathogenic similarities[4]. If we could instead delineate well-defined subtypes that are aligned with underpinning pathobiological changes, we would be able to identify subgroups to which treatment mechanisms do or do not apply and address these long-standing ambiguities. MRI is a strong candidate for data-driven disease classification, because it better reflects the MS pathogenic mechanisms than purely clinical descriptions[5].

Neurodegenerative disorders have a long prodromal period and are lifelong. A key barrier in identifying subtypes of these disorders is to stitch together observations from cross-sectional or longitudinal studies (which are rarely more than a few years long). Grouping individuals on the basis of a similar appearance on a single time-point MRI is not sufficient, as patients belonging to the same subgroup would show different abnormalities as their disease evolves and would appear different. We have recently developed an unsupervised machine learning algorithm, called Subtype and Staging Inference (SuStaIn)[6], to uncover data-driven disease subtypes with distinct temporal progression patterns. This ability to disentangle *temporal* and *phenotypic* heterogeneity makes SuStaIn different from other unsupervised learning or clustering algorithms. The algorithm identifies a set of subtypes in the training data, which can be cross-sectional; each subtype is defined by a pattern of change in a set of features, such as MRI abnormalities. Once the SuStaIn subtypes and their MRI trajectories are identified, the resulting disease model can determine how closely a patient, whose MRI is unseen, belongs to each subtype and stage.

Here, we aimed to redefine subtypes of MS based on a data-driven assessment of the pathological changes visible on MRI scans, rather than the evolution of clinical symptoms, with a view to targeting therapies to subpopulations who share pathogenic mechanisms[7]. In this manuscript, we use phenotype when referring to standard clinical phenotypes (RRMS, PPMS and SPMS) and subtype when referring to MRI-based subtypes, identified using SuStaIn. We applied SuStaIn to a training dataset from previously published clinical trials and observational MRI studies (Table 1) to define subtypes that optimally explained the temporal and phenotypic MRI heterogeneity, and then validated them in an unseen (independent) set (Table 1), thereby confirming the generalisability of the model. Our secondary aim was

**Table 1 Collated datasets.**

| Study name | Population | Design | Participants with eligible MRI | Visits with MRI | Published protocol citation number |
|---|---|---|---|---|---|
| MS datasets in the training dataset** | | | | | |
| Siena | Mixed | Observational | 149 | 595 | 36 |
| Basel | Mixed | Observational | 81 | 239 | 37 |
| DEFINE- CONFIRM, ENDORSE | RRMS | RCT (phase 3) | 1071 | 5208 | 31 |
| OPERA 1 | RRMS | RCT (phase 3) | 801 | 3025 | 32 |
| OPERA 2 | RRMS | RCT (phase 3) | 824 | 3044 | 32 |
| ASCEND | SPMS | RCT (phase 3) | 1002 | 5095 | 25 |
| Lipoic acid | SPMS | RCT (phase 2) | 41 | 111 | 28 |
| MS-STAT 1 | SPMS | RCT (phase 2) | 131 | 373 | 26 |
| MAESTRO 3 | SPMS | RCT (phase 3) | 539 | 1753 | 29 |
| Lamotrigine | SPMS | RCT (phase 2) | 97 | 251 | 27 |
| ARPEGGIO | PPMS | RCT (phase 2) | 409 | 946 | 42 |
| INFORMS | PPMS | RCT (phase 3) | 323 | 758 | 24 |
| PROMISE | PPMS | RCT (phase 3) | 458 | 740 | 23 |
| OLYMPUS | PPMS | RCT (phase 2/3) | 396 | 1630 | 22 |
| MS datasets in the validation dataset** | | | | | |
| CLIMB | RRMS | Observational | 319 | 1950 | 38 |
| ORATORIO | PPMS | RCT (phase 3) | 701 | 2724 | 17 |
| BRAVO | RRMS | RCT (phase 3) | 1,203 | 3009 | 33 |
| MS-SMART | SPMS | RCT (phase 2) | 425 | 1151 | 30 |
| MAESTRO 1 and 2 | SPMS | RCT (phase 3) | 420 | 1570 | 29 |

** We chose training and validation datasets a priori.
Number of patients for each MS phenotypes in the Basel cohort are: 56 RRMS, 19 SPMS, and 6 PPMS and in the Siena cohort are 143 RRMS, 9 PPMS, and 9 SPMS.
Abbreviations: RCT = double-blind randomised controlled trial; RRMS = relapsing-remitting multiple sclerosis; SPMS = secondary progressive multiple sclerosis; PPMS = primary progressive multiple sclerosis.

**Table 2 Patients' characteristics in the training and validation datasets.**

**Clinical phenotypes**

| | Training dataset (N = 6322) | | | Validation dataset (N = 3068) | | |
|---|---|---|---|---|---|---|
| | **RRMS** | **SPMS** | **PPMS** | **RRMS** | **SPMS** | **PPMS** |
| Percentage of population[a] | 46% (2884) | 29% (1837) | 25% (1601) | 49% (1522) | 28% (845) | 23% (701) |
| Age at study entry[b] | 37.44 ± 9.2 | 49.41 ± 8.09 | 49.20 ± 8.41 | 36.53 ± 9.69 | 51.95 ± 7.92 | 44.58 ± 8.02 |
| Female (%) | 68% | 65% | 50% | 69% | 67% | 50% |
| EDSS[c] | 2.5 (1.5–3.5) | 6 (5–6.5) | 4.5 (4–6) | 2.5 (1.5–4.5) | 6 (5–7) | 4.5 (2–7) |
| Disease duration (SD)[b] | 4.62 ± 5.46 | 14.46 ± 8.77 | 4.47 ± 4.56 | 3.24 ± 4.42 | 17.52 ± 9.38 | 2.78 ± 3.1 |
| Progression duration (SD)[b] | — | 5.24 ± 4.04 | — | — | 6.38 ± 5.2 | — |

[a]The total percentages may not add up to 100% because of rounding.
[b]Average years with standard deviation in parentheses.
[c]Median EDSS with interquartile range in parentheses.
Age, disease duration, EDSS and progression duration are calculated at the study entry.
NS non-significant, SD standard deviation, SE standard error of mean, NAWM normal-appearing white matter, EDSS expanded disability status scale, RRMS relapsing remitting multiple sclerosis, SPMS secondary progressive multiple sclerosis, PPMS primary progressive multiple sclerosis.

to determine whether there were differences in progression of disability, disease activity and treatment response between the SuStaIn-derived subtypes at study entry.

## Results

**MRI-based subtypes**. Patient characteristics (age, sex, Expanded Disability Status Scale or EDSS, and disease duration) were similar between the training ($N = 6322$ patients) and the validation datasets ($N = 3068$ patients) (Table 2). Of the 18 MRI features measured, 13 significantly differed between the MS training dataset and control group, and these were retained in the SuStaIn model (see Supplementary Results and Supplementary Fig. 1). Three subtypes, with distinct patterns of evolution, were identified in the training dataset and validated in the validation dataset. On the basis of the earliest MRI abnormality seen in the SuStaIn-defined trajectories, we termed these subtypes cortex-led, normal-appearing white matter (NAWM)-led, and lesion-led (Fig. 1). The pattern of temporal progression for each subtype was characterised as a sequence of stages, each marked by a different combination of changes in regional grey matter volume, NAWM T1/T2 ratio and lesion load (Supplementary Fig. 2). The cortex-led subtype was characterised by an early cortical atrophy in the occipital, parietal, and frontal cortex, which was followed by atrophy in the other grey matter regions and focal T2 lesion accrual and, in the late stage, by a reduction in the T1/T2 ratio (as a proxy for diffuse subtle tissue damage) of the NAWM regions (Fig. 1). The NAWM-led subtype was characterised by an early reduction in the T1/T2 ratio of the cingulate bundle and corpus callosum, followed by a reduction in T1/T2 ratio in the cerebellar, temporal and parietal NAWM, atrophy of the occipital cortex, T2 lesion accrual, and, in the late stage, by atrophy of other cortical regions and the deep grey matter (Fig. 1). The lesion-led subtype was characterised by an early and extensive accrual of T2 lesions, which was followed by early and severe deep grey matter atrophy, atrophy of the occipital, parietal and temporal cortex, and, in the late stage, by a reduction in NAWM T1/T2 ratio (Fig. 1). Staging reflected the pattern of MRI changes within the specific data-driven subtype (Supplementary Fig. 2). The probability that each individual belonged to each of the SuStaIn subtypes is shown in Fig. 2.

When looking at the clinical characteristics of the three subtypes (Table 3), the lesion-led subtype had the highest EDSS, the longest disease duration, the highest lesion load at baseline, the highest lesion accrual over time, and the smallest cortical and deep grey matter volumes at baseline (all $p < 0.001$). Additionally,

the lesion-led subtype showed the highest SuStaIn stage at baseline and the highest annual increase in SuStaIn stage in both the training and validation datasets (Table 3). The most frequent subtype in both the training and validation datasets was the cortex-led subtype, followed by the NAWM-led subtype in the training dataset and the lesion-led subtype in the validation dataset (Table 3). There were no differences in age and sex between the MS subtypes (Table 3).

**Differences in disability progression and disease activity across subtypes**. There was a statistically significant difference in the rate of EDSS progression between the three subtypes in both the training and the validation datasets (log-rank test for three-group comparison, treated and placebo/active comparator patients combined, $p = 0.05$ and $p = 0.006$, respectively, Fig. 3b). In particular, in the training dataset, the lesion-led subtype had 30% higher risk of developing 24-week confirmed disability progression (CDP) than the cortex led subtype (95% confidence intervals (CIs): 5–62%, $p = 0.01$) (Fig. 3a); similarly, in the validation dataset, the lesion-led subtype had 32% higher risk of 24-week CDP than the cortex-led subtype (CIs: 9% to 59%, $p = 0.004$) (Fig. 3b). No other differences in progression of disability between subtypes were seen.

When we investigated the differences in disease activity (relapse rate and enhancing lesions) across subtypes in the training and validation dataset, we found that the lesion-led subtype had the most active disease. When looking at contrast-enhancing lesion counts at the study entry, the lesion-led subtype had the highest number (average count in the training dataset = 2.28, SE = 0.33 and average count in the validation dataset = 2.35, SE = 0.23, $p < 0.001$ for all pairwise comparisons). In the training dataset, the cortex-led subtype had an average contrast-enhancing lesion count of 0.93 (SE = 0.15) and 1.38 (SE = 0.29) in the validation dataset. The NAWM-subtype had an average count of 0.51 (SE = 0.29), $p < 0.001$) in the training dataset and 1.04 (SE = 0.24) in the validation dataset. There was no difference in the numbers of contrast-enhancing lesions between the NAWM-led and the cortex-led subtypes in the training and validation datasets.

Across the three subtypes, the lesion-led subtype had the highest relapse rate in both the training dataset (average = 0.56, SE = 0.07) and validation dataset (average = 0.41, SE = 0.03) (Fig. 4). No other differences in the relapse rate across subtypes were found.

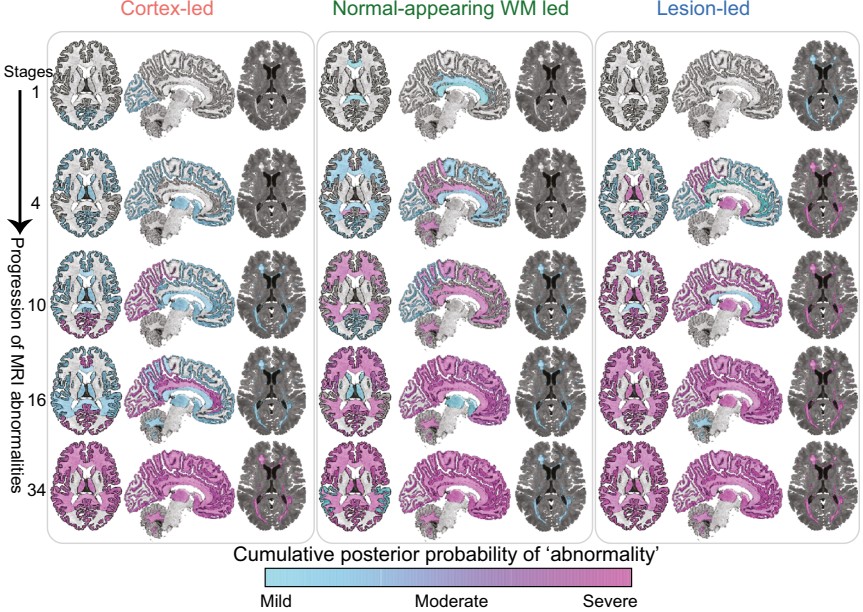

**Fig. 1 MRI-based subtypes.** the evolution of MRI abnormalities in each of the three MRI-based subtypes. The colour shade ranges from blue to pink which represents the probability of abnormality (it can be interpreted as the degree of abnormality) (mild, moderate or severe which approximates 1, 2 and 3 sigma). The cortex-led subtype (left) showed cortical atrophy in the occipital, parietal and frontal cortex in the early stages of the sequences, and a reduction in T1/T2 ratio in the NAWM in the later stages. The normal-appearing white matter (NAWM)-led subtype (middle) showed a reduction in T1/T2 ratio of the cingulate bundle and corpus callosum in the earlier stages of the sequence, and deep grey matter and temporal grey matter atrophy in the later stages. The lesion-led subtype (right) shows early and extensive accumulation of lesions in the earlier stages of the sequence, and a reduction in the T1/T2 ratio in the NAWM in the later stages. The numbers on the left side represent SuStaIn stages. The minimum stage is 1 and the maximum stage is 39 (based on 13 variables that show mild (sigma = 1), moderate (sigma = 2) and severe abnormality (sigma = 3)). Acronyms: NAWM, normal-appearing white matter; SD, standard deviation; GM, grey matter; T1/T2, T1-T2 ratio.

**Patient stratification predicts disability progression**. When we applied SuStaIn to both the training and validation datasets, we found that there were differences in the risk of disability progression between SuStaIn stages. Patients with the highest tertile of stage at baseline (from stage 17 to 39) had the shortest time to 24-week CDP (log-rank $p < 0.0001$); additionally, they showed a 37% higher risk of 24-week CDP (95% CIs: 22–53%) than the patients with the lowest tertile of stage (from stage 1 to 9) and 30% higher risk of 24-week CDP (95% CIs: 17–46%) than patients in the middle tertile group (from stage 10 to 17) (all $p$-values <0.001, see Fig. 5).

There were significant associations between SuStaIn subtypes and stages at baseline with the time-to-24-week CDP (subtypes: overall effect, $\beta = 0.04$, standard error = 0.01, $p = 0.02$; stages: $\beta = -0.06$, standard error = 0.02, $p < 0.001$). However, in the same model, there were no significant associations between the standard clinical phenotypes or baseline EDSS with the time-to-24-week-CDP (phenotypes: overall effect across RRMS, SPMS and PPMS, $\beta = 0.18$, standard error = 0.15, $p = 0.22$), (EDSS: $\beta = 0.02$, standard error = 0.03, $p = 0.26$) suggesting that MRI-based subtypes were more strongly associated with the risk of disability progression than the standard clinical phenotypes.

**Combining MRI-based subtypes with clinical data to predict disease progression**. The concordance index (±standard error) was $0.55 \pm 0.01$ in a survival model with SuStaIn stages and subtypes. The concordance index increased to $0.63 \pm 0.01$ when we added clinical information (EDSS, clinical phenotype, time-walk test and 9-Hole Peg Test) ($p < 0.01$ for performance increase compared to the previous model).

**Differences in treatment response between subtypes**. There were differences in treatment response (defined as the difference in EDSS worsening for each subtype on treatment *vs*. placebo) between the MRI-based subtypes. In particular, the lesion-led subtype showed a significant treatment response in the three phase 3 randomised controlled trials in SPMS and PPMS ($N = 2099$) that were either positive or reported a trend towards a treatment response (Fig. 6). Patients in the lesion-led subtype on active treatment had a significantly slower worsening of EDSS than those on placebo (average percentage difference: −66%, standard error ±25.6%, $p = 0.009$). Similarly, in the pooled analysis of RRMS trials ($N = 2696$), the lesion-led subtype on treatment showed a significant reduction in the rate of EDSS worsening compared to the same subtype in placebo or active comparator arms (−89%, ±44%, $p = 0.04$). No differences in the rate of EDSS worsening were observed between treated patients and those on placebo/active comparator, who belonged to both the NAWM-led and cortex-led subtype.

## Discussion

The application of SuStaIn to a large set of MS MRI scans identified three subtypes, characterised by distinct temporal patterns of MRI changes that could be staged. Results from an independent set corroborated these three subtypes. We found that a patient's baseline subtype and stage was associated with the individual risk of disease progression. Combining MRI-based subtypes with clinical information increased prognostic accuracy when compared with using MRI information alone. The patterns of MRI abnormality in these subtypes provide insights into disease mechanisms and, alongside clinical phenotypes, they may aid stratification of patients for future interventional studies.

Our primary hypothesis was that a model based on MRI rather than solely on clinical data helps to improve a biological

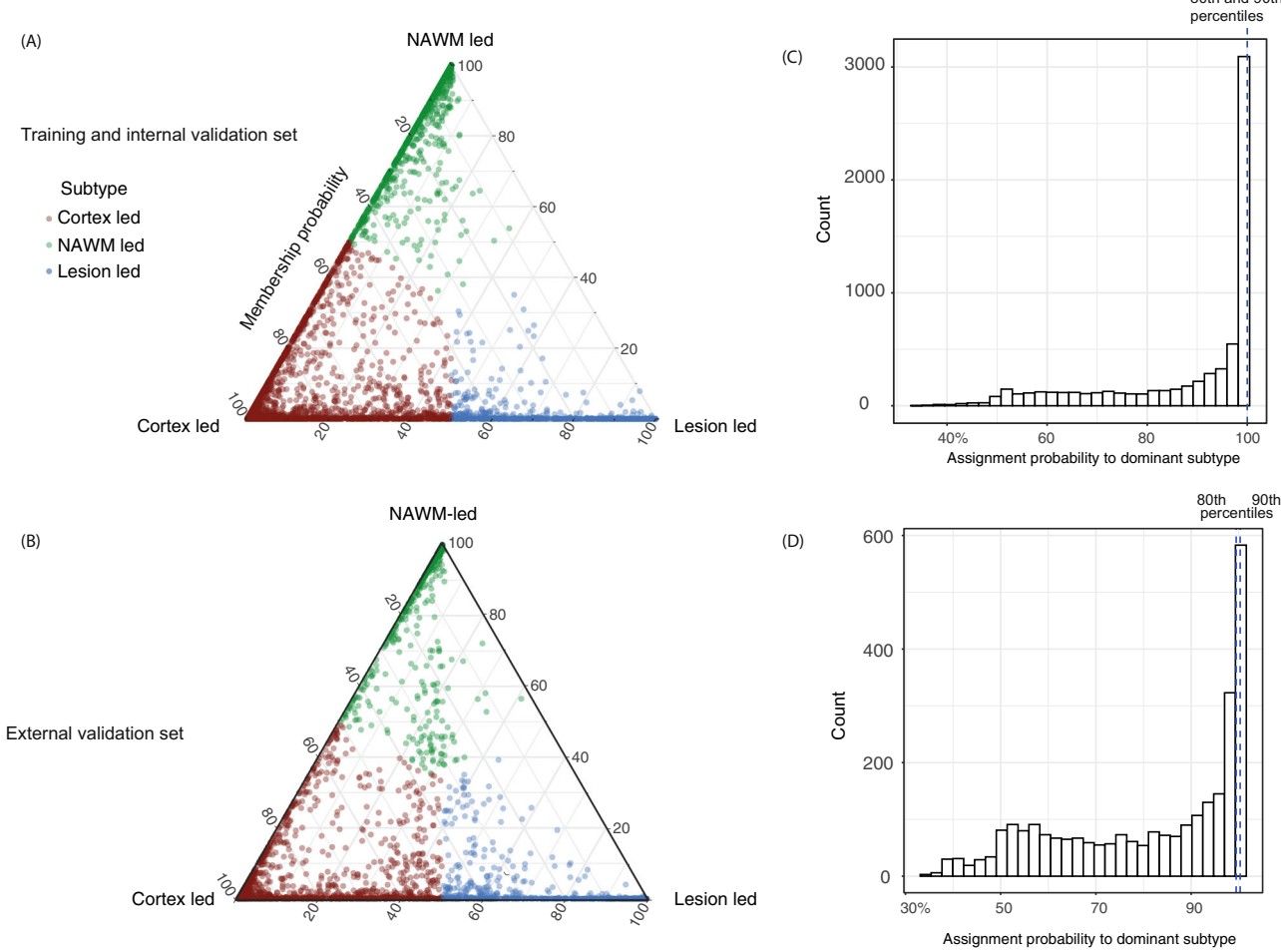

**Fig. 2 Subtype membership in the training, and validation datasets. a** MRI-based subtypes in the training and internal validation dataset and **b** the validation dataset. Assignability of the disease subtype, or membership probability, is shown as the distance from each vertex of the triangle. Each of vertices represent the point at which membership of a given subtype is at its maximum (100%). We assigned each subject to one subtype (shown in red, green and blue) based on their maximum probability. **c**, **d** The 80th and 90th percentiles for the probability of assignment to the dominant subtype was 99.98% and 99.99% (indistinguishable in the figure) in the training dataset. In the validation dataset these percentiles were 99.45% and 99.97% respectively.

**Table 3 Characteristics of MRI-based subtypes in the training and validation datasets.**

**MRI-based subtypes**

| | Training dataset (N = 6322) | | | | Validation dataset (N = 3068) | | | |
|---|---|---|---|---|---|---|---|---|
| | Cortex-led | NAWM-led | Lesion-led | p-Value | Cortex-led | NAWM-led | Lesion-led | p-Value |
| Percentage of population | 43% (2697) | 32% (2011) | 25% (1614) | — | 42% (1279) | 21% (635) | 38% (1154) | — |
| Age ± SD | 43.03 ± 10.28 | 43.70 ± 11.05 | 43.60 ± 10.48 | NS | 41.72 ± 10.43 | 43.13 ± 11.28 | 44.07 ± 11.45 | NS |
| Female (%) | 63% | 63% | 65% | NS | 61% | 72% | 64% | NS |
| EDSS (IQR) | 4.0 (3.5) | 3.5 (3.0) | 4.5 (3.0) | <0.01 | 3.5 (3.5) | 3.5 (3.5) | 4.5 (3) | <0.01 |
| Disease duration | 6.27 (6.92) | 5.56 (6.82) | 9.09 (8.33) | <0.001 | 6.46 ± 8.49 | 7.2 ± 7.88 | 11.63 ± 10.72 | <0.001 |
| Lesion load (SD) | 18.05 (20.20) | 10.51 (12.35) | 47.88 (31.14) | <0.001 | 13.83 (13.8) | 8.81 (8.48) | 39.52 (27.2) | <0.001 |
| Lesion accrual in placebo arms (SE) | 1.02 (±0.31) | 0.88 (±0.41) | 2.64 (±1.50) | <0.01 | 1.57 (0.30) | 0.63 (0.12) | 2.41 (1.51) | <0.001 |
| Brain volume (SE) | 1119 (9.1) | 1092 (1.2) | 1089 (1.15) | <0.001 | 1117 (3.4) | 1120 (6.03) | 1077 (4.82) | <0.001 |
| Cortical volume (SE)* | 453.0 ± 1.7 | 476.60 ± 2.6 | 444.10 ± 2.8 | <0.001 | 455.81 ± 0.52 | 477.32 ± 0.83 | 453.10 ± 0.96 | <0.001 |
| Deep grey matter volume (SE) | 31.16 ± 0.04 | 32.97 ± 0.04 | 28.92 ± 0.07 | <0.001 | 31.47 ± 0.55 | 32.92 ± 0.08 | 29.33 ± 0.08 | <0.001 |
| Baseline SuStaIn stage (SE) | 14.54 ± 0.17 | 13.75 ± 0.13 | 16.15 ± 0.29 | <0.001 | 11.49 ± 0.19 | 10.31 ± 0.27 | 13.82 ± 0.29 | <0.001 |
| SuStaIn stage annual increase in placebo arms (SE) | 0.18 (±0.10) | 0.29 (±0.12) | 0.66 (±0.18) | <0.001 | 0.19 ± 0.21 | 0.12 ± 0.25 | 0.63 ± 0.15 | <0.001 |

*IQR* interquartile range, *NAWM* normal-appearing white matter, *EDSS* expanded disability status scale, *SE* standard error, *SD* standard deviation, *NS* non-significant, *ml* millilitre.

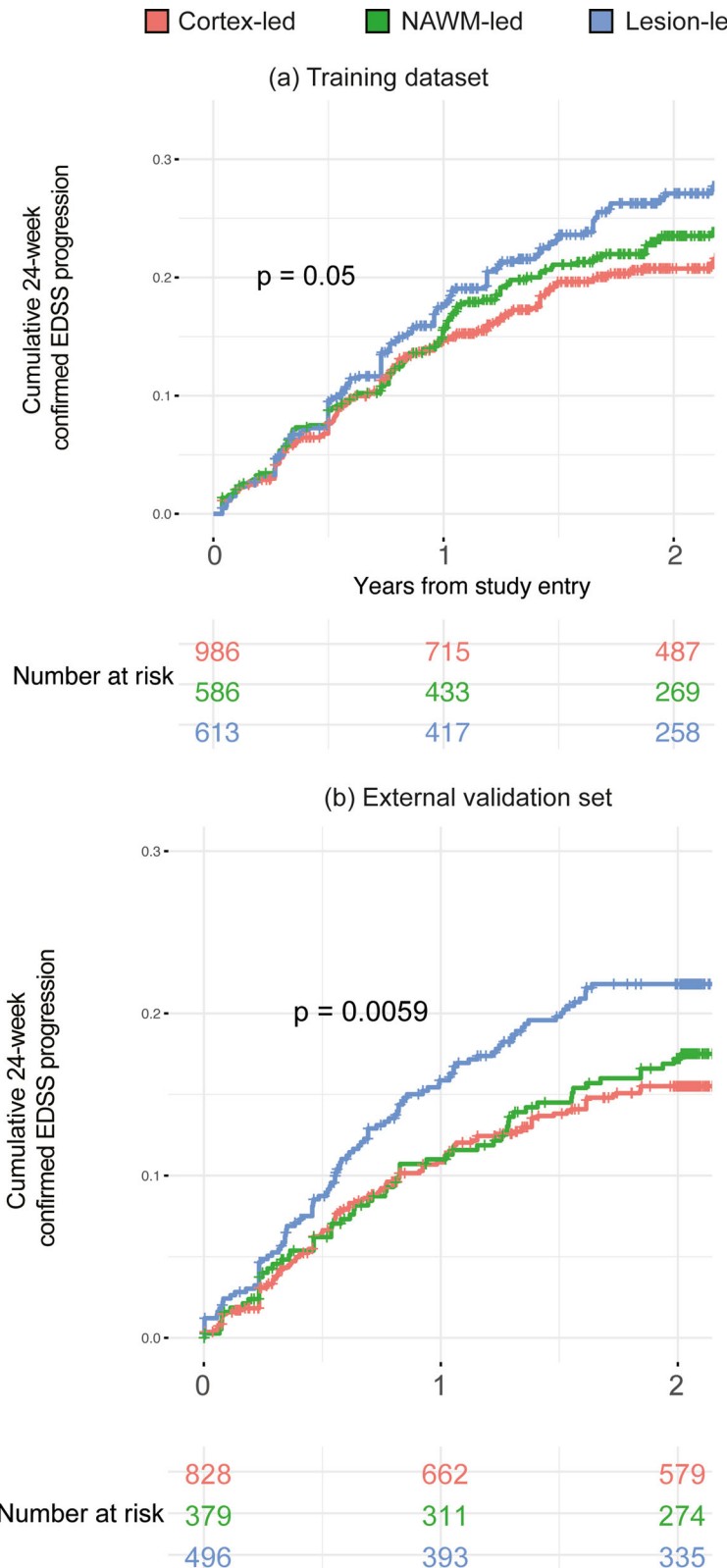

**Fig. 3 MRI-based subtypes and disability progression in the placebo arms.** The lesion-led subtype had a faster EDSS progression than the other two MS subtypes in both the training (**a**) and validation (**b**) sets. Only placebo arms (or comparator arms) of the clinical trials are included. In both **a** and **b** we used the log-rank test, with two-sided *p*-value, to compare survival curves (no correction for multiple comparisons). *EDSS* expanded disability status scale.

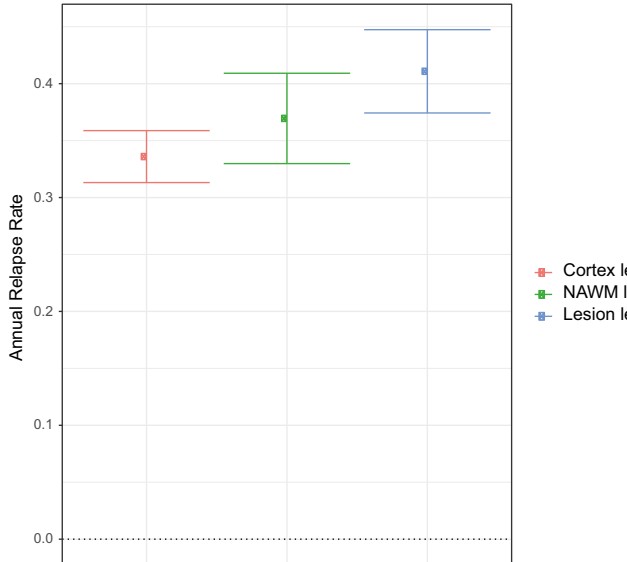

**Fig. 4 MRI-based subtypes predicted disease activity in the placebo arms of the validation dataset.** The average annual relapse rate for each MRI-based subtype. The centre shows the estimated average of annual relapse rate and error bars represent the standard error. The lesion-led subtype had significantly higher annual relapse rate than the cortex-led subtype (*n* = 1663 patients).

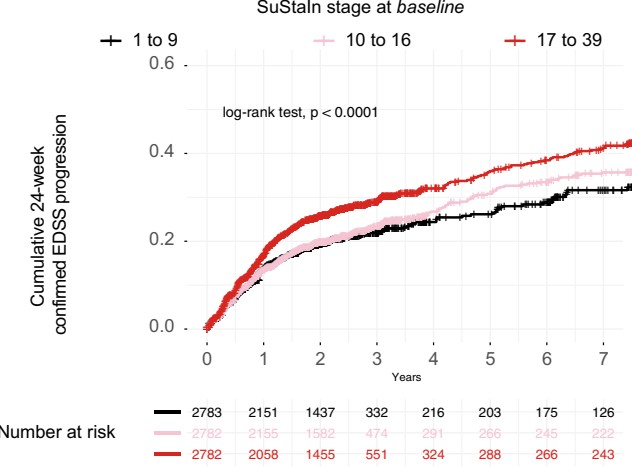

**Fig. 5 Stratification predicts disability progression.** Higher SuStaIn stage at baseline predicted time to disability progression: the higher the stage at baseline, the shorter the time to reach 24-week confirmed EDSS progression. *p*-Value is two-sided (*p* < 0.0001). When we repeated this analysis inside each MRI-based subtype we found similar results (results not shown). *EDSS* expanded disability status scale.

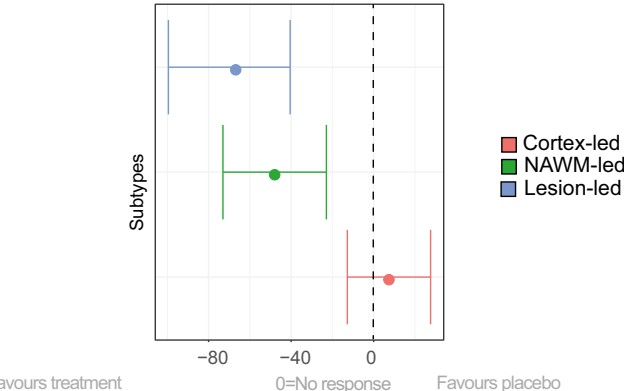

**Fig. 6 Predicting treatment response with MRI-based subtyping in selected RCTs.** Shows the change in EDSS worsening in MRI-based subtypes in the pooled treatment arms of the ORATORIO, ASCEND and OLYMPUS trials (*n* = 2099 patients) compared to the corresponding subtypes in the pooled placebo arms (e.g., lesion-led subtype on treatment *vs*. lesion led subtype on placebo and so forth). Patients in the lesion-led subtype had the largest reduction in the rate of EDSS worsening and were the only group who had a significant treatment response. The circle at the centre of each line represents the model-estimated average of percentage EDSS change. Error bars represent the standard error. Abbreviations: 9HPT, 9-Hole Peg test; NAMW, normal-appearing white matter; EDSS, Expanded Disability Status Scale; RRMS, relapsing-remitting multiple sclerosis; PPMS, primary progressive multiple sclerosis; RCT, randomised controlled trial.

understanding of MS disease progression. There were differences in the risk of disability progression, disease activity and treatment response across subtypes, which suggested that they reflected different pathobiological mechanisms relevant to the manifestations of the disease. Among the three different subtypes, the lesion-led subtype, which was characterised by early and extensive lesion load, showed the fastest accumulation of lesions over time, highest relapse rate and contrast-enhancing lesions, and highest risk of 24-week confirmed disability progression. Interestingly the lesion-led subtype showed the development of deep grey matter atrophy after the T2 lesion accrual. These findings are consistent with substantial focal inflammatory demyelination early on, with

neurodegeneration in the deep grey matter being secondary to lesion accumulation[8,9], related to the degeneration of tracts transected in MS lesions or concomitant inflammatory processes in the white matter and deep grey matter[10,11]. Notably, the lesion-led subtype was the only subtype that showed a significant treatment response in both RRMS and progressive (PP and SP) MS trials, suggesting that SuStaIn, alongside clinical phenotypes, may be used for selecting patients who are more likely to respond to medications targeting inflammatory lesion activity.

When looking at the trajectory of MRI changes defining the cortex-led subtype, this group showed early cortical atrophy in the occipital and parietal regions, subsequent development of atrophy in the other grey matter regions and accumulation of T2 lesions, and late NAWM abnormalities. This suggests that the pathological underpinning of the cortex-led subtype is more insidious and may relate to neurodegeneration in the cortex[8,9] and compartmentalised, chronic inflammation in the white matter, which is not reflected by the visible lesions. The concurrent development of cortical atrophy and accumulation of lesions points to retrograde neurodegeneration of tracts transected in white matter lesions[12]. The majority of patients belonged to the cortex-led subtype, in both the training and validation datasets. This subtype responded to a lesser extent to treatments. This observation suggests that the neurodegenerative component of MS is clinically relevant and remains difficult to target with treatments. It is important to note that SuStaIn disentangles not only the sequence but also the severity of abnormality at each stage. The parietal cortex showed the greatest early atrophy (Supplementary Figure 2) of all cortical regions, regardless of subtype. This might be related to the network mediated pathology in areas of the cortex that were part of the default mode network, such as the parietal lobe (which includes the precuneus) or those connected to the visual system (the occipital cortex) show early damage, as has been demonstrated in previous MRI and histopathological studies[12–14].

When looking at the baseline SuStaIn stage, and change over time, the lesion-led subtype had the highest stage at baseline, and the most rapid increase over time. However, it is important to note that stages are not comparable across subtypes: they represent different patterns of abnormalities, and linear and non-linear changes from one stage to another, which is different across subtypes.

The lesion-led subtypes had the worst prognosis in terms of accumulation of pathology (faster lesion accumulation) and disability progression. Disease duration, both in the training dataset (9.09 years) and the validation dataset (11.63 years), was the highest in the lesion-led subtype. We postulate that this reflects a shorter prodromal period in the lesion-led compared with other subtypes; The higher relapse rate and faster rate of disability progression may contribute to an early diagnosis[15]. Another possibility is that longer disease durations in the lesion-led subtype reflect patients converting from another subtype as they progress. However, this is unlikely to be the case, at least in the time span of a clinical trial, as shown in our reliability analyses, and natural history studies suggest that focal inflammatory activity tends to diminish rather than increase over time[16].

MRI-based subtyping predicted disability progression and treatment response irrespective of clinical phenotypes. The MRI-based subtypes and stages were more strongly associated with EDSS worsening than the baseline EDSS or clinical phenotypes. Even in the validation dataset, where the baseline EDSS ranged from 3.5 to 4.0 for all the three MRI-based subtypes, the MRI-based subtypes still predicted disability progression. We chose a model based on MRI data, rather than including clinical data, to enable a biological understanding of disease progression as opposed to supervised or semi-supervised methods that may include clinical outcomes. Taken together, the MRI-based subtypes can be used alongside clinical phenotypes to add value in prognosticating patient outcomes.

A discrepancy between training and validation datasets was that the second most common subtype in the training dataset was NAWM-led, while in the validation dataset it was lesion-led. This may be explained by the characteristics of PPMS patients of the ORATORIO trial in the validation phase. ORATORIO was a phase 3 trial enriched with patients with active PPMS[17]. This discrepancy highlights that the current phenotypic classification of MS does not fully address the heterogeneity of patients, whereas a data-driven subtyping system can provide a more systematic and objective classification. We could have used individual-based matching as opposed to study-based (clinical trial-based) matching to balance the training and validation dataset, however, we chose the latter to ensure that the treatment response calculations were not biased by changes in the prospectively, and randomly, recruited placebo arms. This choice also enabled us to test our models in truly independent datasets that were acquired by separate investigators at different times.

MS is heterogenous clinically and on MRI, so we included clinical variables (EDSS, 9-Hole Peg Test, and Timed-Walk Test) alongside MRI-based subtypes, to build a more comprehensive model to predict patient outcomes. We found that including clinical assessments at study entry significantly increased the accuracy (concordance index of 0.63 vs. 0.55) of MRI-based subtypes in predicting 24-week CDP. However, we did not include clinical information in the SuStaIn model to subtype patients, because these variables violate monotonicity (variables move in one direction as disease progresses) and normative assumptions (variables are drawn from the sample of healthy volunteers from whom MRI data were acquired) of SuStaIn algorithm[6]. We found that the MRI-based subtypes predicted CDP whilst single MRI variables (lesion load and whole brain volume) did not, suggesting that a comprehensive model is necessary to achieve the difficult task to predict disability progression.

We used data from a large number of clinical trials and observational cohorts. This offered us several advantages, in particular it provided a large dataset, with the ability to robustly assess associations with clinical features and treatment responses, and it meant that our findings can be used in future clinical trials as we did not rely on advanced MRI scanning techniques and the method has proven robust to potential confounding by differences between the many MRI scanners used. However, it also meant that we were not able to use the most sensitive MRI measures, and this was particularly the case for NAWM. Diffusion tensor or magnetisation transfer imaging are more sensitive to intrinsic tissue changes, but these are not routinely collected in phase 3 clinical trials due to standardisation challenges[18]. Since this measure can be affected by the choice of MRI protocol and scanner, we paid special attention to trial effects, centre effects in each trial, and 2D or 3D MRI data acquisition. To mitigate potential differences across scanners we used an internal reference (ventricles) to normalise values (see Supplementary Methods). There were 772 centres in our study; when we looked at the centre or site effects and compared it with the subtype effect across MRI variables (including T1/T2), the MRI measures were more strongly related to subtype than centre. We found similar results when exploring the statistical effect sizes of MRI resolution (2D vs. 3D). To measure and examine trial effects (e.g., recruitment strategy and other confounding factors), we used leave-one-dataset out cross-validation and found excellent consistency across cross-validation folds. Our results were therefore robust to scanner or acquisition effects, which is in line with our previous multi-centre studies showing that grey mattery changes can be quantified from multi-centre studies[19,20]. Our third strategy to show robustness to centre and trial, was applying our model to unseen data. Our model could predict clinical outcomes (EDSS and disease activity) when applied to new centres in unseen data sets, which confirmed that the centre effects are unlikely to significantly affect the predictive performance. However, the output classifications and staging are likely to be noisier than a model trained with data from only a few scanners (bias-variance trade off), which would have been less generalisable.

Our methodology has the potential to be extended to real-world clinical data in future studies by adapting the model (e.g., retraining) to scans acquired in clinical practice and validation in real-world data sets. The spinal cord is affected from early stages in MS and its atrophy is independently associated with disability[21]. However, spinal cord data is not routinely acquired in MS trials and was not available in our study, and it would be interesting in future studies to investigate if spinal cord measures can also independently contribute to SuStaIn subtyping and staging.

In conclusion, we have identified MRI-based subtypes that provide insights into the pathobiological mechanisms of MS and predict disease activity, disability progression, and treatment response better than conventional clinical phenotypes. Our MRI-based subtyping can be undertaken using MRI scans that are already being acquired in clinical trials, and from a single time-point, so it could be prospectively used to enrich future trials with those most likely to respond to treatment, or to subtype patients to specifically look for treatment effects that would otherwise have been overlooked if assessed by clinical MS phenotypes alone.

## Methods

**Study design**. Main goals of our analysis were to identify data-driven subtypes using MRI measures, replicate and test the generalisability of our models in unseen data sets. To perform MRI-based subtyping, we developed a subtyping model by performing the steps summarised in Fig. 7, which included data preparation, MRI

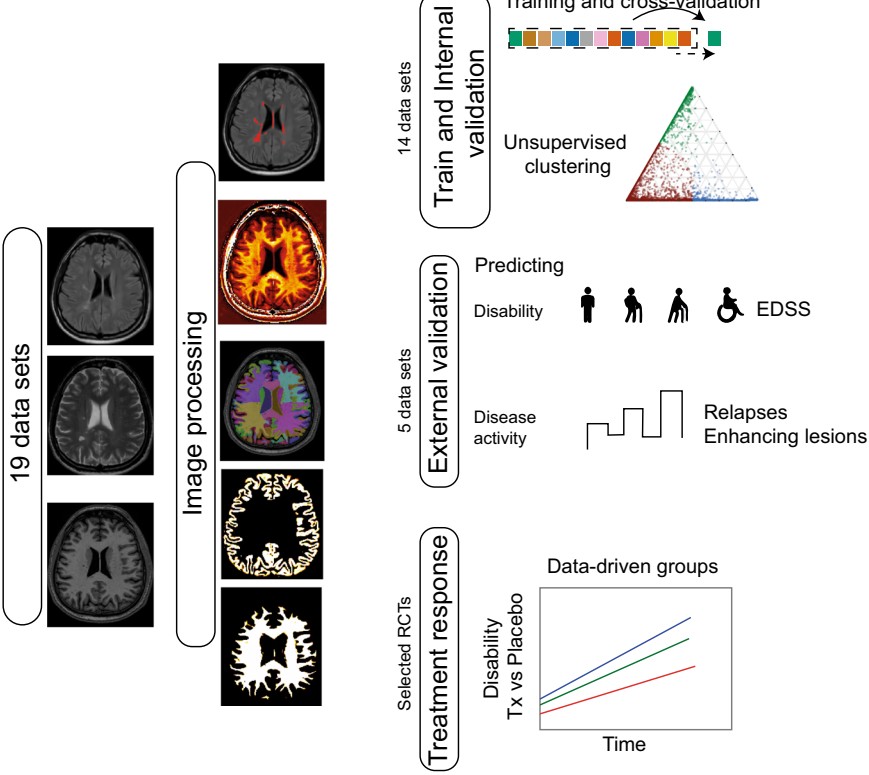

**Fig. 7 Model development and validation.** We processed MRI with a uniform processing pipeline. We trained the model using 14 data sets, and applied the trained model on five independent, unseen data sets for validation. We demonstrated if MRI-based subtyping at baseline, could predict EDSS progression and disease activity. In selected RCTs, from RRMS and progressive (SP and PP MS), we looked at if MRI-based subtyping could predict treatment response. *EDSS* expanded disability status scale, *RCT* randomised controlled trial, *Tx* treatment, *PASAT* paced auditory serial addition test.

processing, extracting variables of interest from processed MRI scans, training an unsupervised machine-learning model and testing it in unseen data sets. We assessed the predictive ability of the SuStaIn subtypes and stages for clinical progression and treatment response, and whether the inclusion of baseline clinical features improved the prediction of clinical progression.

**Participants and clinical outcomes.** We collected clinical and MRI data from 16 MS randomised-controlled trials (RCTs): five trials of PPMS[17,22–24], seven trials of SPMS[25–30], and four trials of RRMS[31–33]; we also included three observational cohorts with mixed MS subtypes[17,22–29,31–38] (Table 1).

Each RCT and observational study had received ethical approval and participants had given written, informed consents at the time of data-acquisition. The Institutional Review Board at the Montreal Neurological Institute (MNI), Quebec, Canada approved this study (Reference number: IRB00010120). The pharmaceutical companies who provided the fully anonymised, individual patient raw data, agreed to pooling data but not re-testing treatment response in individual RCTs. We also included two healthy control data sets: (1) The S1200 Open Access release of the Human Connectome Project, and (2) The UK Biobank data, which were available for download on 1st of February 2019. This project was approved by the UK Biobank (Reference number: 47233).

The Expanded Disability Status Scale (EDSS)[39], which rates neurological impairment, was scored as per individual study protocol. The EDSS was obtained at least one month after a protocol-defined relapse. We defined disability progression confirmed at 24 weeks (or confirmed disability progression (CDP)) as a worsening of EDSS that was sustained on subsequent visits for at least 24 weeks. EDSS progression was defined as a ≥1.5-point increase from a baseline EDSS of 0, a ≥1-point increase from a baseline score of 0.5 to 5.5, and a ≥0.5-point increase from a baseline score greater than 5.5.

**Brain MRI protocol and image processing.** We collected the following brain MRI sequences: T1-weighted, T2-weighted, and Fluid Attenuated Inversion Recovery (T2-FLAIR) MRI (see Supplemental Material for details). We used brain 2D or 3D T1-weighted scans to segment grey and white matter tissues, T2-FLAIR and T1-weighted scans to segment lesions, and T2-weighted scans, together with T1-weighted scans, to obtain T1/T2 ratio. Details of MRI protocols are explained in publications associated with each dataset[17,22–29,31–38].

We applied an identical cross-sectional pipeline (treating each visit independently) to all the visits of patients and healthy controls in which

T1-weighted, FLAIR and T2-weighted MRI were available. We processed scans to obtain the following 18 variables:

> Volumes of the bilateral frontal, parietal, temporal, and occipital grey matter, limbic cortex, cerebellar grey matter and white matter, brainstem, deep grey matter and cerebral white matter
> Volume of total T2 lesions
> Regional T1/T2 ratio of normal-appearing white matter in the corpus callosum, frontal, temporal, parietal, and occipital lobes, cingulate bundle and cerebellum.

Details of image analysis and quality control pipelines are explained in the Supplemental Material. In brief, brain regions were defined based on the Neuromorphometrics atlas (http://www.neuromorphometrics.com). Lesions were segmented using Lesion Segmentation Toolbox[53] and a deep convolutional neural network-based method in DeepMedic[54]. Tissue segmentations were undertaken using T1-weighted volumetric scans processed with Geodesic Information Flows (GIF) software.T1/T2 ratio maps were calculated using a pipeline based on Ganzetti and colleagues method[40], modified to use ventricular CSF rather than the vitreous humour T1/T2 ratios to normalise measures (scan anonymisation and acquisition meant that not all scans included the eyes).

**Statistical analysis**

*Outline.* As mentioned above, SuStaIn is an unsupervised machine learning method that combines disease progression modelling[20] and clustering methods. SuStaIn identifies a set of subtypes with specific patterns in the temporal progression of input variables. What makes SuStaIn unique for modelling neurodegenerative disorders is its ability to model heterogeneity in both the phenotypic and temporal heterogeneity[6], whilst other clustering methods can model one of these two aspects at a time. We first trained and internally validated (cross-validated) SuStaIn in a training dataset, and in the second part we tested it by using an external (unseen and independent) validation dataset. We also investigated the associations between SuStaIn subtypes and stages (and clinical phenotypes) and both disability progression and treatment response by using all the available datasets. In the second part of the analysis, we explored whether there were differences in treatment responses between MRI-based subtypes in three phase 3 RCTs in RRMS and in three phase 3 RCTs in progressive MS.

*Model training and validation.* For each of the 18 MRI variables listed above, we used the two datasets of healthy controls together to fit a Bayesian linear regression

model with the total intracranial volume, sex, age and age squared as independent variables, and each MRI variable as the outcome. We calculated the expected values for each visit using this model and subtracted the observed values to obtain residual values of each MRI variable. We refer to the residual values as adjusted values. We used BAS package version 1.5.3 and R version 3.6.0[41]. We evaluated study and centre effects separately, as explained below in the internal validation and in the Supplemental Methods. We calculated the Z-scores for each MRI variable at each participant's visit by subtracting the adjusted mean value of the healthy volunteers from the adjusted observed value in patients and dividing each patient's MR variable by the standard deviation of the healthy volunteers.

From the 19 datasets available, we a priori chose 14 datasets to create a training dataset, which was used for model training (including cross-validation). These were three phase 3 RRMS trials[31,32], three phase 3 PPMS trials[22–24], two phase 3 SPMS trial[25,29], three phase 2 SPMS trials[26–28], one phase 2 PPMS trial[42] and two observational cohorts[36,37] (see Table 1 and Supplementary Fig. 3 for the complete list). We set aside the remaining four datasets to create a validation dataset, which was used to perform the model testing: one phase 3 RCT in RRMS[33], one phase 3 RCT in PPMS[17], one phase 2 and one phase 3 RCT in SPMS[30], and one observational cohort with mixed MS subtypes[38] (Table 1).

To reduce the dimensionality of the models and computational expenses, we selected the MRI variables, which were found to be 'abnormal' in MS when compared with healthy controls. To do so, we carried out pairwise comparisons between healthy volunteers and patients at their baseline visit and selected the MRI measures whose differences between the groups were associated with a moderate to large effect sizes (>0.5 Cohen's D effect size). We included the volumes for the grey matter areas and the T1/T2 values for the white matter regions, because the T1/T2 and the volume measures were highly correlated for each given region. We entered the MRI variables resulting from the previous step into SuStaIn. Since lower values of volume and T1/T2 ratio are expected to be associated with increased disability, we flipped their signs so that higher Z-scores and estimated stages represented disease worsening.

To find the optimal model (that is, the model with the highest likelihood of explaining MRI variables), we carried out the internal validation with leave-one-dataset-out cross-validation, which allowed us to choose the number of MS subtypes, quantify the uncertainty associated with a given subtype trajectory (or the evolution of MRI abnormalities), and evaluate the stability and robustness of the model across different trials, and MRI protocols (Supplementary Fig. 4). With the leave-one-dataset-out cross-validation procedure, we trained the model on 13 out of the 14 datasets, and evaluated it using the remaining dataset (*held-out sample*). We permuted the training and held-out samples until every dataset was used once as held-out sample, thereby iterating the procedure 14 times.

We selected the best fitting model, which means the model with the optimal number of subtypes, and sequence of MRI abnormality changes in the same subtypes. We started by fitting the SuStaIn model on the 13 training folds with only one subtype and then increased the number of subtypes in steps of one. We calculated the log-likelihood (which expresses predictive accuracy) of each held-out fold for each model and chose the fitted model with the number of subtypes that maximised this log-likelihood. We estimated the uncertainty of the quantification (posterior distribution) using the Markov Chain Monte Carlo (MCMC) algorithm with 100,000 iterations to sample from the posterior distribution of the most likely sequences found in the previous step. To evaluate how consistent MRI-based subtypes were across these 14 datasets in the train and internal validation dataset, we quantified the effects of dataset (and, therefore MRI protocol) on subtype trajectory. We quantified the degree of overlap of posterior distributions of sequences for each subtype across 14 iterations of cross-validation, with the Bhattacharyya coefficient[43] between each pair of subtypes from different folds. The Bhattacharyya coefficient ranges from 0 (no agreement) to 1 (perfect agreement). We calculated all pairwise Bhattacharyya distances across all folds and subtypes pairs for the optimal model and reported the average and standard deviation for each subtype. We fitted our final trained model on all the 14 datasets in training data set to obtain the optimal model to be used for validation.

To compare gender frequencies among MRI-based subtypes and clinical phenotypes, we used Chi-square test. To compare ordinal and continuous outcome variables (e.g., EDSS, SuStaIn stages, age and disease duration), and lesion and cortical volumes we used general linear models (with Poisson distribution for ordinal variables). For longitudinal analysis of lesion volume and cortical volume we used mixed-effects models. In these mixed-effects models, we included hierarchical random effects: the visit variable was nested in subject, and subject variable nested in the dataset variable; we only used the placebo arms of the RCTs to evaluate the natural course of MRI-based subtypes in the absence of treatments and included total intracranial volume, age and sex as fixed-effects, nuisance variables.

*Testing the newly developed model on the validation dataset.* First, we applied SuStaIn to the validation dataset to obtain subtype membership for each subject's visit at baseline (study entry). Secondly, to investigate the differences in the hazard ratio of reaching the 24-week-CDP across the three MRI-based subtypes within each trial, we used Cox regression models. To investigate whether there were differences in disease activity between the MRI-based subtypes, we compared the annual relapse rate in and the number of contrast-enhancing lesions at the study entry using a general linear model[17,33].

*Analyses with merged training and validation datasets.* We used SuStaIn to estimate subtype stages along a trajectory or a sequence. Since there were 13 variables with three Z-scores each, each subtype included 39 stages, which ranged from one (the earliest stage) to 39 (the last stage). To investigate whether the MRI-based subtypes and the standard clinical MS phenotypes were associated with disability progression, we constructed a mixed-effects model. In this model, time to reach 24-week-confirmed EDSS progression was the outcome variable and trial was a random-effects variable. Fixed-effects predictors were MRI-based subtypes and stages at baseline, standard MS phenotypes, age, sex, and EDSS at baseline. We performed additional analyses to test the reliability and stability of the SuStaIn subtypes over time in both the train and test set (see Supplemental Material for details). To compare the strength of associations of clinical MS phenotypes and MRI-based subtypes with time to 24-week CDP, we used the standardised time-to-event variables and compared the average effect sizes across the clinical phenotypes and the MRI-based phenotypes using a general linear model.

*Combining clinical variables with the MRI-based subtypes to predict disease progression.* We used EDSS, Timed-Walk Test, and 9-Hole Peg Test performance at study entry in addition to the MRI-based subtypes to predict time to 24-week CDP. We used a Cox-regression model in which the time to 24-week CDP was the outcome variable and MRI-subtypes, together with the other variables, including clinical scores, were independent variables. To provide a measure of predictive performance of our model at the individual level, we calculated the concordance index, which is the fraction of pairs where the observation with the higher survival time had a higher probability of survival, as predicted by our model. It ranges from 0 for no concurrence to 1 for perfect prediction. Since clinical information violates SuStaIn assumptions of monotonicity and normative distributions in healthy controls (explained in the Discussion), we did not include them in the SuStaIn algorithm.

*Difference in treatment response across MRI-based subtypes.* We explored whether there were differences in treatment responses across the MRI-based subtypes, by looking at the rate (or slope) of EDSS worsening in three phase 3 RCTs in progressive MS (ORATORIO, ASCEND and OLYMPUS[22,25,44]) pooled together, and in three phase 3 RCTs in RRMS (DEFINE-CONFIRM-ENDORSE, OPERA1 and OPERA2) also merged together. We chose these trials because they were either positive trials[17,31,32] or had a subgroup that showed a trend towards a treatment response in previous publications[22,25]. For the RRMS trials, we merged the arms with different doses of the experimental drug, included the placebo arms, and excluded the active comparator arms. We used a linear mixed-effects model in which EDSS was the outcome variable with group, time, and group x time interaction as the independent variables. Group was a binary variable indicating either a given subtype on treatment or the same subtype on placebo. To adjust for repeated measures and correlated residual errors, we added hierarchical random effects to our model, in which visits were nested in the subject variable. We reported the difference in percentage change of EDSS worsening between groups, which we refer to as treatment response throughout this manuscript. We used NLME package version 3.1 and Survival package version 2.44 inside R version 3.6.0 for statistical analysis[45,46]. In all the statistical analyses in which we reported p-values, we used two-tailed tests.

**Reporting summary.** Further information on research design is available in the Nature Research Reporting Summary linked to this article.

## Data availability
Data from patients are controlled by pharmaceutical companies (publications with this information are listed in references in the Table 1) and therefore are not publicly available. Request to access data should be forwarded to data controllers via the corresponding author. Processed data can be requested by qualified investigators from the corresponding author. Data from healthy volunteers (UK Biobank and Human Connectome Project) are publicly available and can be requested by application through the UK Biobank and Human Connectome Project websites (https://www.ukbiobank.ac.uk/ and https://www.humanconnectome.org/).

## Code availability
SuStaIn code is available publicly at https://github.com/ucl-pond/pySuStaIn. We used the code at commit 54b92b154acc9d8757751edea50d1fcfab672015.

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

## Acknowledgements

This study was supported by the International Progressive MS Alliance (IPMSA, award reference number PA-1603-08175). We are grateful to all the IPMSA investigators who have contributed trial data to this study as part of EPITOME: Enhancing Power of Intervention Trials Through Optimized MRI Endpoints network (see the list of investigators in the appendix). This study was also supported by the National Institute for Health Research University College London Hospitals Biomedical Research Centre. O.C. is a National Institute of Health Research (NIHR) Professor (grant code: RP-2017-08-ST2-004). We are grateful to Professor Geraint Rees for his comments. We thank Rozie Arnaoutellis, Istvan Morocz, and Caramanos Zografos for coordinating and organising this study. We thank Jonathan Steel for IT support during this work. This research in part has been conducted using the UK Biobank Resource under Application Number 47233. Data have also been provided in part by the Human Connectome Project, WU-Minn Consortium (Principal Investigators: David Van Essen and Kamil Ugurbil; 1U54MH091657) funded by the 16 NIH Institutes and Centers that support the NIH Blueprint for Neuroscience Research; and by the McDonnell Center for Systems Neuroscience at Washington University. D.C.A. has received funding for this work from Engineering and Physical Sciences Research Council Grants M020533, M006093, and J020990. This project has received funding from the European Union's Horizon 2020 Research and Innovation Programme under Grant Agreements 666992. MS-SMART is an investigator-led project sponsored by University College London (UCL). This project (reference 11/30/11) is funded by the Efficacy and Mechanism Evaluation (EME) Programme, a Medical Research Council (MRC) and National Institute for Health Research (NIHR) partnership. The views expressed in this publication are those of the author(s) and not necessarily those of the MRC, NIHR, or the Department of Health and Social Care. A.J.T. is a National Institute of Health Research (NIHR) Emeritus Senior Investigator. A.L.Y. is supported by an MRC skills development fellowship (MR/T027800/1).

## Author contributions

A.E., O.C., D.C., A.Y., P.W. and D.C.A. designed the study. O.C., F.B., D.C., D.L.A., S.N., D.C.A. and A.J.T. have supervised this study. A.E., A.Y., F.P. and P.W. have designed the experiments. D.A., S.N. and C.R.G.G. acquired data. All authors reviewed, contributed to drafting and approved the manuscript.

## Competing interests

A.E. has received speaker's honoraria from Biogen and At The Limits educational programme. He has received travel support from the National Multiple Sclerosis Society and honorarium from the Journal of Neurology, Neurosurgy and Psychiatry for Editorial Commentaries. In the last 3 years D.C. has received honoraria from Excemed (2017) for faculty-led education work; had meeting expenses funded by the IMSCOGS (2019), EAN

(2018), ECTRIMS (2018) and Société des Neurosciences (2017). He is a consultant for Biogen and Hoffmann-La Roche. He has received research funding from the International Progressive MS Alliance, the MS Society, and the National Institute for Health Research (NIHR) University College London Hospitals (UCLH) Biomedical Research Centre. He is a member of the MS Society's Biomedical Grant Review Panel and a trustee of the MS Trust. O.C. has received research grants from the MS Society of Great Britain & Northern Ireland, National Institute for Health Research (NIHR) University College London Hospitals Biomedical Research Centre, EUH2020, Spinal Cord Research Foundation, and Rosetrees Trust. She serves as a consultant for Novartis, Teva, and Roche and has received an honorarium from the American Academy of Neurology as Associate Editor of Neurology and serves on the Editorial Board of Multiple Sclerosis Journal. CRGG has received research grants form Sanofi and the National Multiple Sclerosis Society. F.B. has received compensation for consulting services and/or speaking activities from Bayer Schering Pharma, Biogen Idec, Merck Serono, Novartis, Genzyme, Synthon BV, Roche, Teva, Jansen research and IXICO and is supported by the NIHR Biomedical Research Centre at UCLH. A.J.T. has received honoraria/support for travel for consultancy from Eisai, Hoffman La Roche, Almirall, and Excemed, and support for travel for consultancy as chair of the International Progressive MS Alliance Scientific Steering Committee, and from the National MS Society (USA) as a member of the Research Programs Advisory Committee. He receives an honorarium from SAGE Publishers as Editor-in-Chief of Multiple Sclerosis. Journal and a free subscription from Elsevier as a board member for the Lancet Neurology. D.L.A. has received research grant funding and/ or personal compensation for consulting from Acorda, Adelphi, Alkermes, Biogen, Celgene, Frequency Therapeutics, Genentech, Genzyme, Hoffman-La Roche, Immuene Tolerance Network, Immunotec, MedDay, EMD Serono, Novartis, Pfizer, Receptos, Roche, Sanofi-Aventis, Canadian Institutes of Health Research, MS Society of Canada, and International Progressive MS Alliance; and holds an equity interest in NeuroRx Research. F.B., D.C.A. and A.E. hold equity stake in Queen Square Analytics. S.N. has received research funding from the Canadian Institutes of Health Research, the International Progressive MS Alliance, the Myelin Repair Foundation and Immunotec. He has received honoraria/travel support from Genentech and MedDay, and personal compensation from NeuroRx Research. The remaining authors declare no competing interests.
