## [Peer Review File · Nature Communications]

Reviewers' Comments:

Reviewer #1:

Remarks to the Author:

Although the majority of the issues have been addressed in the revised version of the manuscript, there are still several concerns on this study:

1. Since MS is a heterogeneous disease, it should be relevant to include in the SuStaIn algorithm also clinical variables of MS patients, in order to possibly model both the pathological processes (visible on MRI) and clinical outcomes. This would probably result in a more comprehensive model that better describes the heterogeneity of MS subtypes
2. How image segmentation results from both 2D and 3D T1-weighted sequences could be combined? Segmentation accuracies from these two sequences are hardly comparable
3. Regarding comment #9, in addition to the criteria of eligibility on the presence of T1-weighted and T2-weighted sequences, how have been managed the occurrence of artefacts or other confounding factors on images?
4. Regarding comment #13, the authors stated that the Ganzetti et al. method has been implemented for the T1/T2 ratio maps estimation. However, in Ganzetti et al. intensity normalization has not been performed using ventricle values. This intensity normalization is not optimal, since also cerebro-spinal fluid values could be affected by the MS pathology
5. There are still a lot of Supplemental materials that are not so 'supplementary', but most of them are required to better understand all the analyses performed. Methods are still presented in a fragmentary way, making the logical thread very hard to follow. The authors should discuss how the findings on MS subtypes could replace or be combined with clinical phenotypes in a clinical setting. Moreover, the prognostic values of the SuStaIn model in comparison to the single MRI variables is still uncertain.

Reviewer #2:

Remarks to the Author:

I was one of the previous reviewers (Reviewer 2 in the authors' response) and the majority of this review will focus on the changes made by the authors since the first submission.

The authors have made substantive changes to the manuscript, which have greatly increased clarity of the potential value of the findings. However, some of my previous concerns remain unaddressed, and while the manuscript has value beyond these concerns, I would like to see the main contributions revised and additional clarification on the limitations of the study. To provide some context, in my previous review, I had written:

"Using 13 features to define 3 subtypes seems like a drastic oversimplification of MS disease processes. This is apparent when examining Supplementary Figure 4, which shows that the 3-subtype model is actually very similar to the 2-subtype model. A more comprehensive model would include a larger suite of imaging features and clinical phenotypes in combination, so that pathological processes and clinical evolution can be modeled jointly, perhaps in a semisupervised framework. The clinical phenotypes should be used as part of the prediction model in a complementary way, rather than used as independent comparators."

The authors' latest results show that changing the test data set to match the clinical phenotypes in the training set produced much more consistent clustering results in the MRI subtypes, which is further confirmation that the clinical phenotypes and imaging features should be modelled together for clinical prediction. The authors partially addressed this by adding this clarifying statement to the discussion section: "Taking together these findings with the treatment effect identified in the lesion-led subtype, we conclude that the MRI-derived subtypes can be used alongside clinical phenotypes to add value in prognosticating patient outcomes. We chose a model based on MRI data, rather than including clinical data, to enable a biological understanding of disease

progression as opposed to supervised or semi-supervised methods that may include clinical outcomes." I think this perspective is fine, but necessitates revising the main claims of the manuscript (including in the abstract). Rather than proposing that MRI subtypes can be used on their own for stratifying patients for clinical trials, which is premature based on the current evidence, the main claims should be:

-A model based on MRI rather than including clinical data helps to improve a biological understanding of MS disease progression.

-The MRI-derived subtypes can be used alongside clinical phenotypes to add value in prognosticating patient outcomes.

In addition, given that machine learning is a core technology in the methodology, readers will want to know from the outset that 13 features selected based on statistical group differences to non-MS volunteers were used to define the 3 subtypes, which I also recommend stating in the abstract.

Finally, the limitation in the methodology for choosing the number of subtypes should also be explicitly discussed. Supplementary Figure 4 shows that the 3-subtype model is actually very similar to the 2-subtype model, and this is especially apparent when recognizing that the studies with the largest differences in CVIC between 2 and 3 subtypes also had the largest absolute values in CVIC, so the relative differences were small.

Reviewer #3:

Remarks to the Author:

I believe the authors have markedly improved the manuscript. All of my questions were answered comprehensively and, as far as I can judge, the other reviewer's questions were similarly addressed. The work is highly original and I am sure will be a landmark paper in its field.

Helmut Butzkueven

Reviewer Comments

Reviewer #1

1. Although the majority of the issues have been addressed in the revised version of the manuscript, there are still several concerns on this study:

Since MS is a heterogeneous disease, it should be relevant to include in the SuStaln algorithm also clinical variables of MS patients, in order to possibly model both the pathological processes (visible on MRI) and clinical outcomes. This would probably result in a more comprehensive model that better describes the heterogeneity of MS subtypes.

Response: We thank the reviewer for this comment. Clinical data violate monotonicity (variables must move in one direction as disease progresses) and normative assumptions (MRI and clinical variables are drawn from a sample of healthy volunteers) of the SuStaln algorithm (Young et al., 2018, *Nature Communications*) and so cannot be included in the subtyping model in its current form. However, we were able to include clinical data alongside the SuStaln subtype in a new model predicting disease progression, and as the reviewer anticipated this has improved predictive power. We have made the following changes:

Methods, Page 25 and 26

“Combining clinical variables with the MRI-derived subtypes to predict disease progression

We used EDSS, Timed-Walk Test, and 9-Hole Peg Test performance at study entry in addition to the MRI-derived subtypes to predict time to 24-week CDP. We used a Cox-regression model in which the time to 24-week CDP was the outcome variable and MRI-subtypes, together with the other variables, including clinical scores, were the independent variables. To provide a measure of predictive performance of our model at the individual level, we calculated the concordance index, which is the fraction of pairs where the observation with the higher survival time had a higher probability of survival, as

predicted by our model. It ranges from 0 for no concurrence to 1 for perfect prediction.

Since clinical information violates SuStaln assumptions of monotonicity and normative distributions in healthy controls (explained in the Discussion), we could not include them in the SuStaln algorithm.”

Results, Page 9

*“Combining MRI-derived subtypes with clinical data to predict disease progression
The concordance index (\pm standard error) was 0.55 ± 0.01 with SuStaln stages and subtypes. The concordance index increased to 0.63 ± 0.01 when we added clinical information (EDSS, clinical phenotype, timed-walk test and 9-Hole Peg Test) ($p < 0.01$ for performance increase compared to the previous model). “*

Discussion, Page 14

“MS is heterogenous clinically and on MRI, so we included clinical variables (EDSS, 9-Hole Peg Test, and Timed-Walk Test) alongside MRI-derived subtypes, to build a more comprehensive model that predicts patient outcomes. We found that including clinical assessments at study entry significantly increased the accuracy (concordance index of 0.63 compared with 0.55 without clinical measures) of MRI-derived subtypes in predicting 24-week CDP. However, we did not include clinical information in the SuStaln model to subtype patients, because these variables violate monotonicity (variables move in one direction as disease progresses) and normative assumptions (variables are drawn from the sample of healthy volunteers from whom MRI data were acquired) of SuStaln algorithm⁶.”

2. How image segmentation results from both 2D and 3D T1-weighted sequences could be combined? Segmentation accuracies from these two sequences are hardly comparable.

We thank the reviewer for highlighting this important point. We determined if scan type (2D or 3D) or SuStaln types had greater influence on regional brain volumes. We constructed

mixed-effects models in which data acquisition (2D or 3D) and MRI-derived SuStaln subtypes were tested as predictors of regional brain volumes and found that the effect size of the 2D or 3D acquisition was smaller than that of the MRI-derived subtypes (ranging from 1.5 times to 9.9 times depending on the brain region). This is in line with our previous studies on different data sets, in which we found that the effects of mixing 2D and 3D MRI protocols were sufficiently smaller than differences in MS phenotypes (Eshaghi et al, 2018, *Brain* and Eshaghi et al. 2018, *Annals of Neurology*).

Supplemental Methods, Page 51

“...We also determined whether 2D versus 3D MRI protocols, or SuStaln model subtypes explain the majority of variation in regional brain tissue volumes. ... To compare the effects of 3D vs 2D MRI data acquisition, we used a mixed effects model, in which data acquisition (2D or 3D) and MRI-derived subtypes were predictors and regional brain volumes were outcomes. We quantified the effect size for each variable and compared them using general linear model contrasts with the multcomp package in R.”

Supplemental Results, Page 53

“... When we compared the effects of 2D or 3D acquisition of T1-weighted MRI vs MRI-derived subtypes on brain volumes, the magnitude of effect sizes was, 1.5 times (for the parietal lobe) and 9.9 times (for the deep grey matter) larger for MRI-derived subtypes than 2D or 3D MRI acquisitions.”

Discussion, Page 16

“... Our results were therefore robust to scanner or acquisition effects, which is in line with our previous multi-centre studies showing that grey matter changes can be quantified from multi-centre studies^{18,19.}”

3. Regarding comment #9, in addition to the criteria of eligibility on the presence of T1-weighted and T2-weighted sequences, how have been managed the occurrence of artefacts or other confounding factors on images?

We have added the details of how we have managed the occurrence of artefacts or other confounding factors on MRI scans (including T1 and T2-weighted sequences).

Supplemental Methods, Page 47 and 48

“Quality assurance data was provided by sponsors of each clinical trial, and we excluded MRI data where a quality issue (e.g., acquisition artefact) was flagged. AE also reviewed all scans to ensure coverage of the whole brain, including the cerebellum, and confirm that there were no additional visible image artefacts that could affect scan processing.

4. Regarding comment #13, the authors stated that the Ganzetti et al. method has been implemented for the T1/T2 ratio maps estimation. However, in Ganzetti et al. intensity normalization has not been performed using ventricle values. This intensity normalization is not optimal, since also cerebro-spinal fluid values could be affected by the MS pathology.

Thank you for prompting us to clarify this. We could not use masks of eye and temporal muscles (as suggested by Ganzetti et al.) because in several clinical trials anonymisation of the scans removed the face and eyes, or temporal muscles were not included in the MRI acquisition protocols. Therefore, we used the ventricular CSF for normalisation. We agree that CSF values could be affected by MS pathology, although the dominant signal will be from water, but measurements in the temporal muscles (and vitreous humour) may also be affected by MS, as Ganzetti et al. have highlighted.

In order to address this reviewer's concern, we compared T1/T2 ratios in the CSF between MS patients and controls and found no significant difference. The value of T1/T2 in the ventricles had a similar mean value in both groups (normalized against population mean: -0.0045 in healthy controls and MS). We have added more information to the manuscript to clarify this important point.

Methods, Page 49

".. Ganzetti et al. used measurements of the vitreous humour and temporal muscles to normalise T1/T2 ratios; however, because of data anonymisation, subjects' eyes were removed from the scans in several of the clinical trials included in this study. Therefore, we used T1/T2 ratio of the ventricular CSF to normalise individual T1/T2 ratios. When we compared ventricular CSF T1/T2 ratios between MS patients and controls, no significant differences were detected."

- 1. There are still a lot of Supplemental materials that are not so 'supplementary', but most of them are required to better understand all the analyses performed.**

We agree with this comment. We have moved the following sections to the main manuscript:

Methods, Page 19

"We applied an identical cross-sectional pipeline (treating each visit independently) to all the visits of patients and healthy controls in which T1-weighted, FLAIR and T2-weighted MRI were available..."

Methods, Page 20

“In brief, brain regions were defined based on the Neuromorphometrics atlas (<http://www.neuromorphometrics.com>). Lesions were segmented using Lesion Segmentation Toolbox⁵³ and a deep convolutional neural network-based method in DeepMedic⁵⁴. Tissue segmentations were undertaken using T1-weighted volumetric scans processed with Geodesic Information Flows (GIF) software. T1/T2 ratio maps were calculated using a pipeline based on Ganzetti and colleagues method⁶⁹, which was modified to use ventricular CSF rather than the vitreous humour T1/T2 ratios to normalise individual measures (scan anonymisation meant that not all scans included the eyes).”

2. Methods are still presented in a fragmentary way, making the logical thread very hard to follow.

We have tried to make it less fragmented and explain the logical thread by:

- 1) Adding a paragraph at the beginning of Methods that summarises the study design (Methods, Page 18)
- 2) Reducing the number of subheadings and new paragraphs (Methods, Page 20 to 24)
- 3) Merged several fragmented paragraphs into single paragraphs (Methods Page, 20 to 24)
- 4) Moving some of the Supplemental Material parts, which were felt to be essential, back into the main manuscript (see Reviewer 1's Comment 5 above).

7.The authors should discuss how the findings on MS subtypes could replace or be combined with clinical phenotypes in a clinical setting.

We agree and have added the following paragraph to discuss this important point:

Discussion, Page 16

“Our methodology has the potential to be extended to real-world clinical data in future studies by adapting the model (e.g., retraining) to scans acquired in clinical practice and validation in real-world data sets.”

8. Moreover, the prognostic values of the SuStaln model in comparison to the single MRI variables is still uncertain.

We agree and have included additional analyses in the supplementary materials, and have added a comment to the discussion as follows:

Supplementary results, Page 52

“Comparison across MRI subtypes and established imaging measures in predicting CDP

The MRI-derived subtypes were significantly associated with the time to 24-week CDP while the lesion volumes or brain volumes were not (in the same model): MRI-derived subtypes (hazard ratio (HR)=1.15, SE=0.06, p=0.04), total brain volume (HR=1.13, SE=0.07, p=0.1), and lesion load (HR=1.08, SE=0.07, p=0.25).”

Discussion, Page 15

“We found that the MRI-derived subtypes predicted CDP whilst single MRI variables (lesion load and whole brain volume) did not, suggesting that a comprehensive model is necessary to achieve the difficult task to predict disability progression.”

Reviewer #2

I was one of the previous reviewers (Reviewer 2 in the authors’ response) and the majority of this review will focus on the changes made by the authors since the first submission. The authors have made substantive changes to the manuscript, which have greatly increased clarity of the potential value of the findings. However, some of my previous concerns remain unaddressed, and while the manuscript has value

beyond these concerns, I would like to see the main contributions revised and additional clarification on the limitations of the study.

We thank the reviewer. We have revised the main contributions and added clarification of the limitations of our study in our responses below.

1. To provide some context, in my previous review, I had written:

“Using 13 features to define 3 subtypes seems like a drastic oversimplification of MS disease processes. This is apparent when examining Supplementary Figure 4, which shows that the 3-subtype model is actually very similar to the 2-subtype model. A more comprehensive model would include a larger suite of imaging features and clinical phenotypes in combination, so that pathological processes and clinical evolution can be modeled jointly, perhaps in a semisupervised framework. The clinical phenotypes should be used as part of the prediction model in a complementary way, rather than used as independent comparators.”

The authors’ latest results show that changing the test data set to match the clinical phenotypes in the training set produced much more consistent clustering results in the MRI subtypes, which is further confirmation that the clinical phenotypes and imaging features should be modelled together for clinical prediction. The authors partially addressed this by adding this clarifying statement to the discussion section:

“Taking together these findings with the treatment effect identified in the lesion-led subtype, we conclude that the MRI-derived subtypes can be used alongside clinical phenotypes to add value in prognosticating patient outcomes. We chose a model based on MRI data, rather than including clinical data, to enable a biological understanding of disease progression as opposed to supervised or semi-supervised methods that may include clinical outcomes.” I think this perspective is fine, but necessitates revising the main claims of the manuscript (including in the abstract).

Rather than proposing that MRI subtypes can be used on their own for stratifying

patients for clinical trials, which is premature based on the current evidence, the main claims should be:

-A model based on MRI rather than including clinical data helps to improve a biological understanding of MS disease progression.

-The MRI-derived subtypes can be used alongside clinical phenotypes to add value in prognosticating patient outcomes.

We thank the reviewer for these comments. We have revised main claims of the study to address this important point:

Abstract, Conclusion

“The pattern of abnormality in MRI-derived subtypes provides novel insights into the pathobiological disease mechanisms by showing the main changes that contribute to each subtype. The MRI-derived SuStaln MS subtyping helps to improve biological understanding of MS. MRI-derived subtypes alongside clinical phenotypes could be used to stratify patients for enrolment into clinical trials.”

Discussion, Page 11

“... The patterns of MRI abnormality in these subtypes provide novel insights into disease mechanisms and, alongside clinical phenotypes, they could aid stratification of patients for clinical trials and healthcare interventions.”

Discussion, Page 12

“...suggesting that SuStaln, alongside clinical phenotypes, could be used for selecting patients who are more likely to respond medication targeting inflammatory lesion activity.”

2. In addition, given that machine learning is a core technology in the methodology, readers will want to know from the outset that 13 features selected based on statistical group differences to non-MS volunteers were used to define the 3 subtypes, which I also recommend stating in the abstract.

We agree with the reviewer. We have made the following changes to the abstract and restated in the beginning of the Results section.

Abstract

“Out of 18 MRI metrics analysed, 13 significantly differed between the MS training set and control group (n=14,928), and they were retained in the SuStaln model. On the basis of the earliest observed MRI abnormality, amongst these 13 variables, these subtypes were named cortex-led, normal-appearing white matter-led, and lesion-led.”

Results, Page 6

“Of the 18 MRI features measured, 13 significantly differed between the MS training set and control group, and these were retained in the SuStaln model (see Supplementary Results).”

3. Finally, the limitation in the methodology for choosing the number of subtypes should also be explicitly discussed. Supplementary Figure 4 shows that the 3-subtype model is actually very similar to the 2-subtype model, and this is especially apparent when recognizing that the studies with the largest differences in CVIC between 2 and 3 subtypes also had the largest absolute values in CVIC, so the relative differences were small.

We thank the reviewer for highlighting this. Absolute values of the CVIC are not shown in the figure because they cannot be compared between trials. This is because trials have different

sample sizes (similar to other model comparison measures such as BIC or AIC whose

absolute values do not have an interpretation). For this reason, the vertical axis of Supplemental Figure 3 is the *relative* change with respect to the one subtype model in the same subtype (rather than *absolute* values, which are not shown). Larger starting values on the vertical axis, therefore, imply a larger change in model likelihood of two-subtype model vs one-subtype model. A relative difference of 6 is considered strong evidence (Young et al 2018, *Nature Communications*) in these models. Average difference was 147 when comparing three subtype model with two-subtype model; therefore, the evidence for three subtype model was very strong.

We have added this information to the Supplemental Figure 3:

Supplementary Figure 3 Legend, Page 57

“The vertical axis shows the change in CVIC with respect to the one-subtype model. The relative average difference between three-subtype model and two-subtype model was 147. A relative difference of 6 is considered strong evidence that a model is better than another (Young et al, 2018).”

Reviewer #3

I believe the authors have markedly improved the manuscript. All of my questions were answered comprehensively and, as far as I can judge, the other reviewer's questions were similarly addressed. The work is highly original, and I am sure will be a landmark paper in its field. Helmut Butzkueven

We thank Prof Butzkueven for his previous review and supportive comment.

Reviewers' Comments:

Reviewer #1:

Remarks to the Author:

The authors have addressed all my comments.

Reviewer #2:

Remarks to the Author:

The authors have done a very nice job addressing my last remaining concerns. I especially appreciate the clarification on Supplementary Figure 3, which helps to put the methodology on a more solid footing. I recommend acceptance of the manuscript.